# Entomopathogenic nematodes increase predation success by inducing cadaver volatiles that attract healthy herbivores

Xi Zhang[1], Ricardo AR Machado[1], Cong Van Doan[1], Carla CM Arce[2], Lingfei Hu[1], Christelle AM Robert[1]*

[1]Institute of Plant Sciences, University of Bern, Bern, Switzerland; [2]Institute of Biology, University of Neuchatel, Neuchatel, Switzerland

**Abstract** Herbivore natural enemies protect plants by regulating herbivore populations. Whether they can alter the behavior of their prey to increase predation success is unknown. We investigate if and how infection by the entomopathogenic nematode *Heterorhabditis bacteriophora* changes the behavior of healthy larvae of the western corn rootworm (*Diabrotica virgifera*), a major pest of maize. We found that nematode-infected rootworm cadavers are attractive to rootworm larvae, and that this behavior increases nematode reproductive success. Nematode-infected rootworms release distinct volatile bouquets, including the unusual volatile butylated hydroxytoluene (BHT). BHT alone attracts rootworms, and increases nematode reproductive success. A screen of different nematode and herbivore species shows that attraction of healthy hosts to nematode-infected cadavers is widespread and likely involves species-specific volatile cues. This study reveals a new facet of the biology of herbivore natural enemies that boosts their predation success by increasing the probability of host encounters.

DOI: https://doi.org/10.7554/eLife.46668.001

*For correspondence: christelle.robert@ips.unibe.ch

**Competing interests:** The authors declare that no competing interests exist.

## Introduction

Herbivore natural enemies such as predators, parasites and parasitoids play a key role in terrestrial ecosystems by reducing herbivore abundance (*Vidal and Murphy, 2018*). Biological control relies on this form of top-down control to protect crops from herbivores (*Mills, 2001*). In order to exert their effects, herbivore natural enemies need to make contact with their hosts. Natural enemies have evolved various behavioral strategies to maximize their chances for host encounters (*Jackson and Pollard, 1996*; *Olberg et al., 2000*; *de Rijk et al., 2013*; *de Rijk et al., 2016*). Predators and parasitoids for instance can use herbivore-induced plant volatiles to locate herbivores (*Turlings et al., 1990*). Herbivores on the other hand can detect and actively avoid contact with natural enemies (*Lima and Dill, 1990*). The interplay between behavioral adaptations of herbivores and natural enemies is likely to be an important determinant for the success of herbivore natural enemies and their capacity to suppress herbivore pests.

A key step in the life of many herbivore natural enemies is the acquisition of new hosts once the old host is exploited. Predators and parasitoids acquire new hosts through hunting, ambushing and trapping their prey. Parasites with indirect life cycles can also facilitate the transfer to new hosts through host manipulation strategies, including changes in color, smell and behavior of their current hosts to attract alternate hosts (*Mauck et al., 2010*; *Thomas et al., 2011*; *Daoust et al., 2015*; *Bakker et al., 2017*). How parasites with direct life cycles (i.e. involving a single host) can facilitate transmission from exploited hosts to new healthy hosts is less well established (*Adamo et al., 2014*). Recent studies show that insect bacterial pathogens can induce to changes in volatile emissions in infected individuals, which results in the attraction of non-infected individuals (*Keesey et al., 2017*).

Whether predators, parasitoids and multicellular parasites with direct life cycles can use volatiles to attract additional hosts or prey remains to be determined. Furthermore, whether behavioral manipulation of herbivores by natural enemies can enhance top-down control of herbivores is unclear.

Entomopathogenic nematodes (EPNs) are obligate pathogens of soil insects and important biological control agents of insect herbivores (*Grewal et al., 2005*; *Pilz et al., 2014*). Infective juveniles (IJs) are the only EPN stage surviving in the soil looking for a host. IJs can locate a host using volatile cues that are emitted by herbivores and herbivore-infested plants (*Rivera et al., 2017*). Once an IJ comes into contact with a compatible host, it penetrates it and injects entomopathogenic symbiotic bacteria. EPNs from the Steinernematidae and Heterorhabditidae families inoculate bacteria of the genus *Xenorhabdus and Photorhabdus* respectively, which kill the insect by septicemia and toxemia (*Duchaud et al., 2003*). The EPNs then feed on bacteria and infected host-tissues and multiplies within the cadaver. Eventually, a new generation of IJs emerges from the cadaver and begins searching for new hosts. A crucial factor that determines the success of EPNs is their transmission efficiency from exploited to healthy hosts (*Labaude and Griffin, 2018*). As EPNs are much less mobile than their hosts, they may have evolved host manipulation strategies to increase the probability of host encounters. Conversely, herbivores may effectively avoid EPNs by detecting their presence. So far, the impact of EPNs on host behavior has not been studied in detail.

Here, we investigated the interactions between healthy and EPN-infected conspecific insects. We first studied the behavior of the western corn rootworm (WCR, *Diabrotica virgifera virgifera*), a major root pest of maize that occurs sympatrically with *H. bacteriophora* and is the target of EPN-based biological control programs. WCR can use plant toxins to repel *H. bacteriophora* (*Robert et al., 2017*), but successful biological control of WCR through *H. bacteriophora* has been reported (*Toepfer et al., 2005*; *Toepfer et al., 2008*). Through a series of behavioral experiments, we demonstrate that healthy WCR larvae are attracted to EPN-infected cadavers. We identify a volatile that is specifically released from infected cadavers and is sufficient to attract WCR larvae and increase WCR control by EPNs. Finally, we determine the impact of EPN infection on attraction of different insect and EPN species to test whether this phenomenon is widespread. Collectively, these experiments reveal a novel facet of EPN biology that enhances their capacity to infect and kill insect herbivores.

## Results

### Western corn rootworm larvae are attracted to nematode-infected cadavers

To explore how WCR responds to the presence of nematode-infested conspecifics, we infested root-feeding WCR larvae with *H. bacteriophora*. We then measured the attractiveness of the maize +WCR+EPN complexes at different time points over 96 hr using belowground olfactometers. Maize +WCR+EPN complexes were attractive to WCR 48 hr after WCR and EPN application (*Figure 1A*). The attraction coincided with high root consumption by WCR (*Figure 1B*) and high production of the WCR attractant (*E*)-β-caryophyllene (*Robert et al., 2013*; *Robert et al., 2012a*) by the attacked maize roots (*Figure 1C*). At this time point, approx. 30% of the WCR larvae were infected with EPNs (*Figure 1D*). The attractive effect of maize-WCR-EPN complexes disappeared at 72 hr but reappeared at 96 hr (*Figure 1A*). The increased recruitment of WCR 96 hr post infestation was unexpected, as at this time point 95% of WCR larvae were infected and killed by EPNs (*Figure 1D*), and root removal and root (*E*)-β-caryophyllene emissions had decreased markedly (*Figure 1B and C*).

These experiments show that the interaction between maize, WCR and EPNs results in dynamic changes in WCR recruitment over time, with maize+WCR+EPN complexes becoming attractive as WCR infection by EPNs progresses.

To better understand the factors that render plant-herbivore-nematode complexes more attractive to WCR 96 hr post infection, we quantified WCR recruitment to WCR or EPNs individually and in combination. Plants in the presence of WCR or EPNs alone were not attractive to WCR than plants alone at this time point. However, plants in the presence of WCR under EPN attack were attractive to WCR larvae (*Figure 1E*). We next tested whether the cadavers of EPN-infected WCR larvae attract WCR by putting infected and uninfected WCR cadavers into small filter paper cages and burying them beneath individual maize plants. WCR larvae preferred EPN-infected WCR cadavers

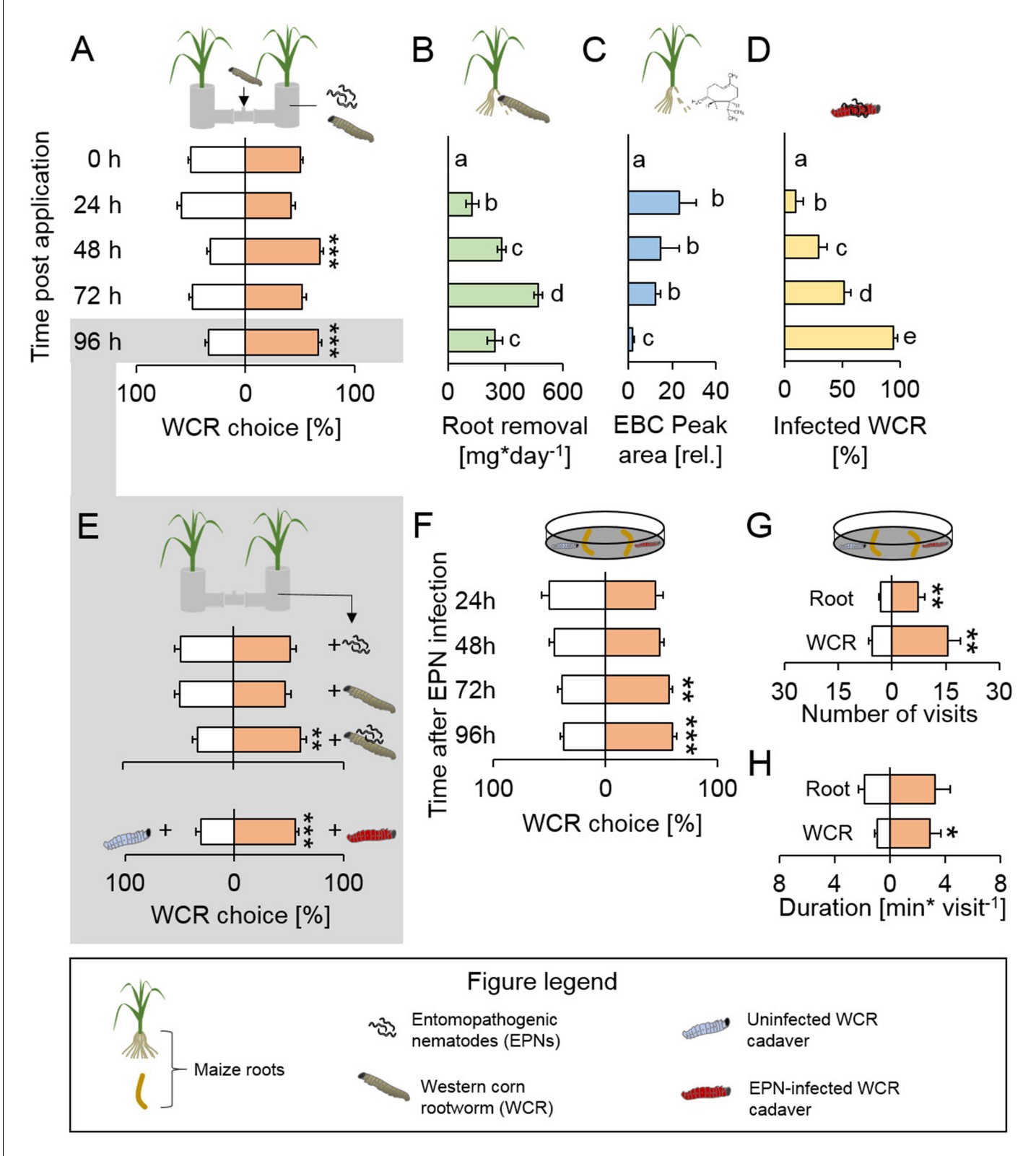

**Figure 1.** Root herbivore recruitment dynamics of plant-herbivore-natural enemy complexes reveal that herbivore cadavers infected by entomopathogenic nematodes attract healthy herbivores. (**A**) Proportions (mean ± SEM) of western corn rootworm (WCR) choosing between healthy maize plants and maize plants infected with conspecifics and entomopathogenic nematodes (EPNs) in belowground olfactometers. WCR choice was measured 0 hr, 24 hr, 48 hr and 96 hr after infection (n = 45). (**B**) Root removal (mean ± SEM) by WCR larvae 0 hr, 24 hr, 48 hr and 96 hr after infection

*Figure 1 continued on next page*

*Figure 1 continued*

(n = 5–8). (C) (E)-β-caryophyllene (EBC) production (mean ± SEM) of maize roots 0 hr, 24 hr, 48 hr and 96 hr after infection (n = 3–5). (D) WCR infection by EPNs (mean ± SEM) 0 hr, 24 hr, 48 hr and 96 hr after infection (n = 8). (E) Proportions (mean ± SEM) of WCR larvae choosing healthy plants or plant +WCR+EPN complexes (n = 20), healthy plants or WCR-infested plants (n = 20), healthy plants or plant+WCR+EPN complexes (n = 20), caged uninfected or EPN-infected WCR cadaver (n = 33). Larval preference was assessed in belowground olfactometers 96 hr post infection. (F) Proportion of WCR larvae orienting towards uninfected and EPN-infected WCR cadavers in the presence of maize root pieces in petri dish assays (n = 10–15) (G-H) Number and duration of visits (mean ± SEM) of WCR larvae exposed to uninfected and EPN-infected WCR cadavers in the presence of maize root pieces (n = 6). Stars indicate significant differences based on analysis of variance (*: p<0.05, **: p<0.01, ***: p<0.001). Raw data are available in *Figure 1—source data 1*.

DOI: https://doi.org/10.7554/eLife.46668.002

The following source data and figure supplement are available for figure 1:

**Source data 1.** Raw data for *Figure 1*.
DOI: https://doi.org/10.7554/eLife.46668.004
**Figure supplement 1.** Attraction of the western corn rootworm to nematode-infected cadavers requires plant background odors.
DOI: https://doi.org/10.7554/eLife.46668.003

over uninfected cadavers (*Figure 1E*). Time course analysis revealed that the attraction to EPN-infected cadavers was strongest 96 hr after infection, shortly before the emergence of IJs (*Figure 1F*). To better understand how WCR larvae respond to the presence of EPN-infected cadavers, we performed additional behavioral experiments in petri dishes using EPN-infected and uninfected WCR cadavers and small root pieces to provide plant background odors. WCR larvae visited infected cadavers more often than uninfected cadavers (*Figure 1G*) and spent more time per visit on infected cadavers (*Figure 1H*). The number of root visits was also increased for roots that were close to infected cadavers (*Figure 1G*). WCR larvae did not show any preference for uninfected or EPN-infected cadavers in the absence of plant roots (*Figure 1—figure supplement 1A*). We hypothesized that this may either be due to plant background odors which are required to elicit WCR search behavior, or due to plant-mediated attraction, where EPN-infected WCR cadavers render roots more attractive to WCR. To test the second hypothesis, we exposed maize roots to uninfected and EPN-infected cadavers, removed the cadavers and evaluated WCR choice. WCR larvae did not show any preference for the different roots (*Figure 1—figure supplement 1B*). Together, these experiments demonstrate that EPN infection of WCR directly increases volatile-mediated recruitment and cadaver contact of healthy foraging WCR larvae.

## The presence of nematode-infected cadavers increases WCR recruitment and nematode predation success in the soil

To investigate whether the presence of EPN-infected cadavers increases EPN predation by attracting healthy herbivores, we conducted experiments in soil arenas (*Figure 2A*). IJs were added together with frozen-killed control larvae or flash-frozen EPN-infected cadavers to the two sides of the arenas, and healthy living WCR larvae were released in the center of the arenas (*Figure 2A*). Cadavers were flash-frozen to avoid confounding effects of EPNs that may have otherwise emerged from the cadavers. Two times more WCR larvae were recovered in the vicinity of EPN-infected cadavers (*Figure 2B*). Furthermore, EPN infection rates were higher on the side where EPN-infected cadavers were buried (*Figure 2C*). The number of emerging EPN juveniles was similar on both sides (*Figure 2D*). Overall, three times more EPN offspring emerged on the side with EPN-infected cadavers (*Figure 2E*).

## Nematode-infection induces volatile release from herbivore cadavers

To identify possible volatile cues that may attract WCR to EPN-infected cadavers, we performed headspace analyses of infected an uninfected WCR cadavers. GC-FID analysis revealed no significant difference in $CO_2$ emissions between uninfected and EPN-infected WCR cadavers (*Figure 3—figure supplement 1*). Headspace solid phase micro extraction (SPME) and GC-MS followed by automated alignment and peak picking revealed 279 distinct volatile features, including 15 features that were exclusively detected in the headspace of infected cadavers (*Figure 3A*). Principal component analysis (PCA) revealed a clear separation of volatile profiles from infected and uninfected cadavers along PC axis 1 (*Figure 3B*). A single volatile eluding at 18.23 min explained 45.6% of the variability of axis

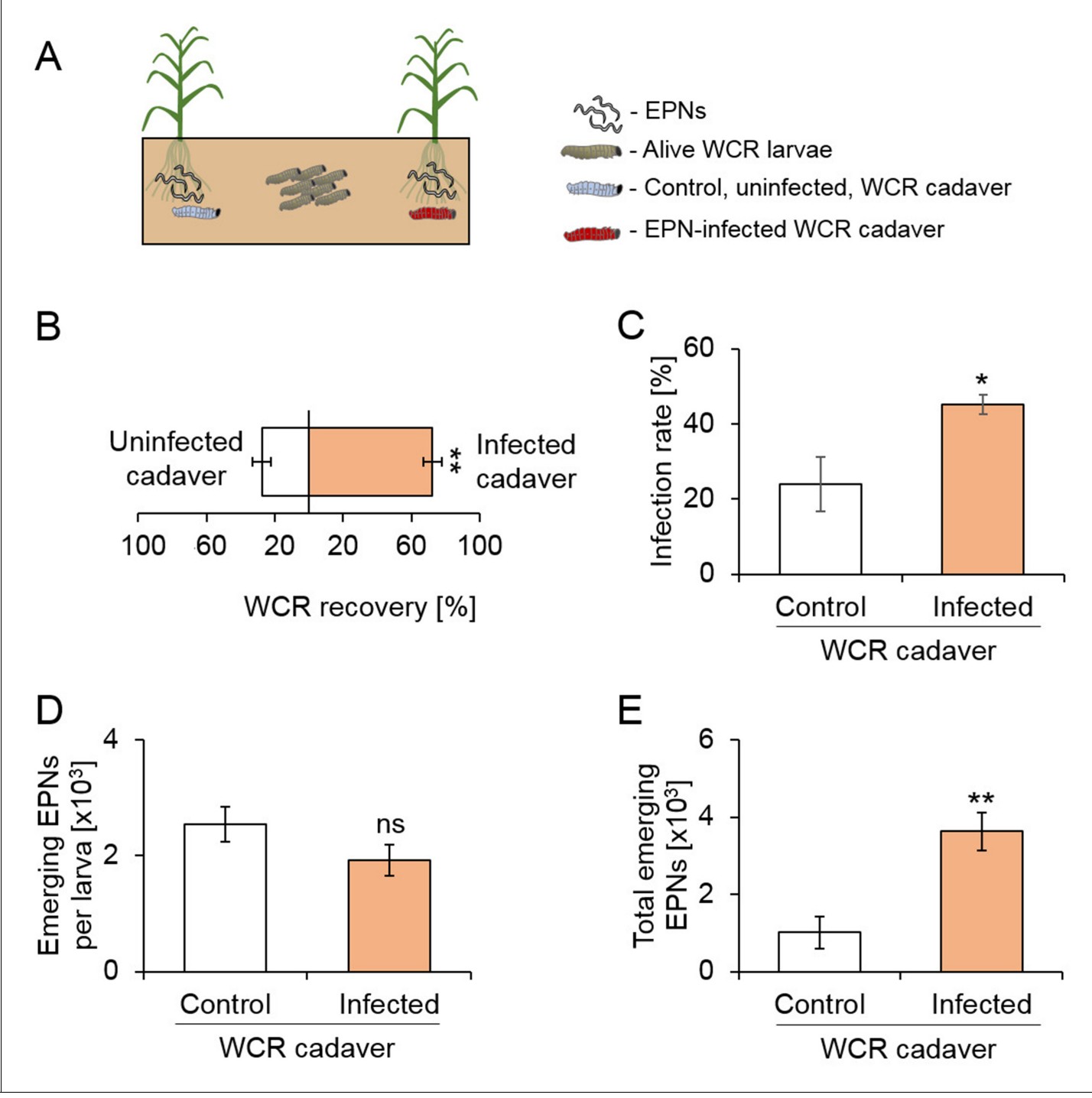

**Figure 2.** The presence of EPN-infected cadavers increases herbivore recruitment, nematode predation success and offspring production. (**A**) Visual representation of experimental setup. Entomopathogenic nematodes (EPNs) were applied on both sides of the arenas, in presence of a healthy or infected WCR cadavers. Eight western corn rootworm (WCR) larvae were then released in the middle and recollected after five days (n = 9). (**B**) Proportions (Mean ± SEM) of WCR larvae recovered from each side (n = 9). (**C**) WCR infection rates (Mean ± SEM) on each side (n = 9). (**D**) Number (Mean ± SEM) of emerging EPN juveniles per infected WCR larva ($n_{Control}$ = 4, $n_{infected}$ = 9). (**E**) Total number of EPNs (Mean ± SEM) emerging from the WCR larvae on each side (n = 9). Stars indicate significant differences based on analysis of variance (**: $p < 0.01$, ***: $p < 0.001$). Ns: non-significant. Raw data are available in *Figure 2—source data 1*.

DOI: https://doi.org/10.7554/eLife.46668.005

The following source data and figure supplement are available for figure 2:

*Figure 2 continued on next page*

*Figure 2 continued*

**Source data 1.** Raw data for *Figure 2*.
DOI: https://doi.org/10.7554/eLife.46668.007
**Figure supplement 1.** Picture of the adapted white-traps used in the study.
DOI: https://doi.org/10.7554/eLife.46668.006

one and was exclusively present in the headspace of infected cadavers (*Figure 3C*). Additional manual integration and relative quantification of the 11 highest peaks in the headspace chromatograms revealed that 9 out of 11 integrated peaks were emitted in higher quantities by infected cadavers (*Figure 3D*). The highest peak, at retention time 18.23 min, was only present in chromatograms of infected cadavers. Based on comparisons of retention times, mass spectra and co-injection of a synthetic standard, the compound was identified as butylated hydroxytoluene (BHT). No BHT was detected in blank controls and in the headspace of different materials that were used for rearing and experimentation. Thus, environmental contamination as a source of BHT can be excluded. A single infected WCR cadaver was found to release up to 5 ng of BHT per hr (*Figure 3—figure supplement 2A*). Emission kinetics showed that BHT starts being released 72 hr after EPN infection (*Figure 3—figure supplement 2A*), which corresponds to the onset of increased WCR recruitment to EPN-infected cadavers (*Figure 1—figure supplement 1*). Therefore, further investigations focused on BHT as a potential volatile attractant of WCR. BHT has originally been described as a synthetic antioxidant (*Babu and Wu, 2008*), but has also been identified in the headspace of cyanobacteria, algae, and fungal pathogens (*Babu and Wu, 2008*; *Hussaini et al., 2011*). To test whether the EPN endosymbiotic bacterium *Photorhabdus laumondii* subsp. *laumondii* (*Machado et al., 2018*) may be responsible for BHT production, we injected it into WCR larvae directly, which resulted in visual infection symptoms and mortality similar to EPN infection. No BHT release from *P. laumondii* infected WCR cadavers was detected (*Figure 3—figure supplement 2B*). *P. laumondii* grown in vitro did not release any BHT either (*Figure 3—figure supplement 2B*). We also did not detect any BHT release from EPNs (*Figure 3—figure supplement 2B*) or uninfected WCR cadavers (*Figure 3*). Instead, BHT was exclusively detected in EPN-infected WCR cadavers (*Figure 3—figure supplement 2B*). These results imply that BHT release is specific to infection of WCR by EPNs.

## Butylated hydroxytoluene emitted by nematode-infected cadavers attracts healthy hosts and increases nematode infection success

Based on the correlation between BHT release and WCR attraction (*Figure 1F* and *Figure 3—figure supplement 2*), we hypothesized that BHT may recruit WCR larvae to EPN-infected cadavers. At physiologically relevant doses, synthetic BHT was attractive to WCR and elicited responses that were comparable to infected cadavers (*Figure 4A*). Furthermore, BHT complementation of uninfected WCR cadavers rendered them as attractive as infected cadavers (*Figure 4A*). Thus, BHT is sufficient to attract WCR to nematode infected cadavers. BHT exposure may not only increase the recruitment of herbivore hosts but may also affect nematode success. To test this hypothesis, we pre-incubated WCR larvae with BHT and measured nematode infection rates in a no-choice setup. Measuring BHT-exposed WCR larvae headspace after rinsing them revealed low traces of BHT ($<0.001$ ng.WCR$^{-1}$.h$^{-1}$, *Figure 4—figure supplement 1A*). Pre-exposure of WCR to BHT increased infection rates by 19% (*Figure 4B*). The increased infection rate was not due to increased attraction of EPNs, as the amounts of BHT released by BHT-exposed larvae was not sufficient to trigger a preference choice (*Figure 4—figure supplement 1B*). By contrast, pre-exposure of EPNs to BHT did not affect their capacity to infect WCR (*Figure 4D*). Interestingly, EPNs were attracted by EPN-infected cadavers as well as BHT similar to WCR (*Figure 4C*). Thus, in addition to attracting WCR larvae, BHT increases EPN infection success and attracts EPNs themselves.

## Butylated hydroxytoluene increases nematode predation success in the soil

To investigate the impact of BHT release by EPN-infected cadavers on EPN predation success, we performed experiments in soil arenas as described above. IJs were added with or without BHT to different sides of the arenas, and healthy WCR larvae were released in the middle (*Figure 5A*).

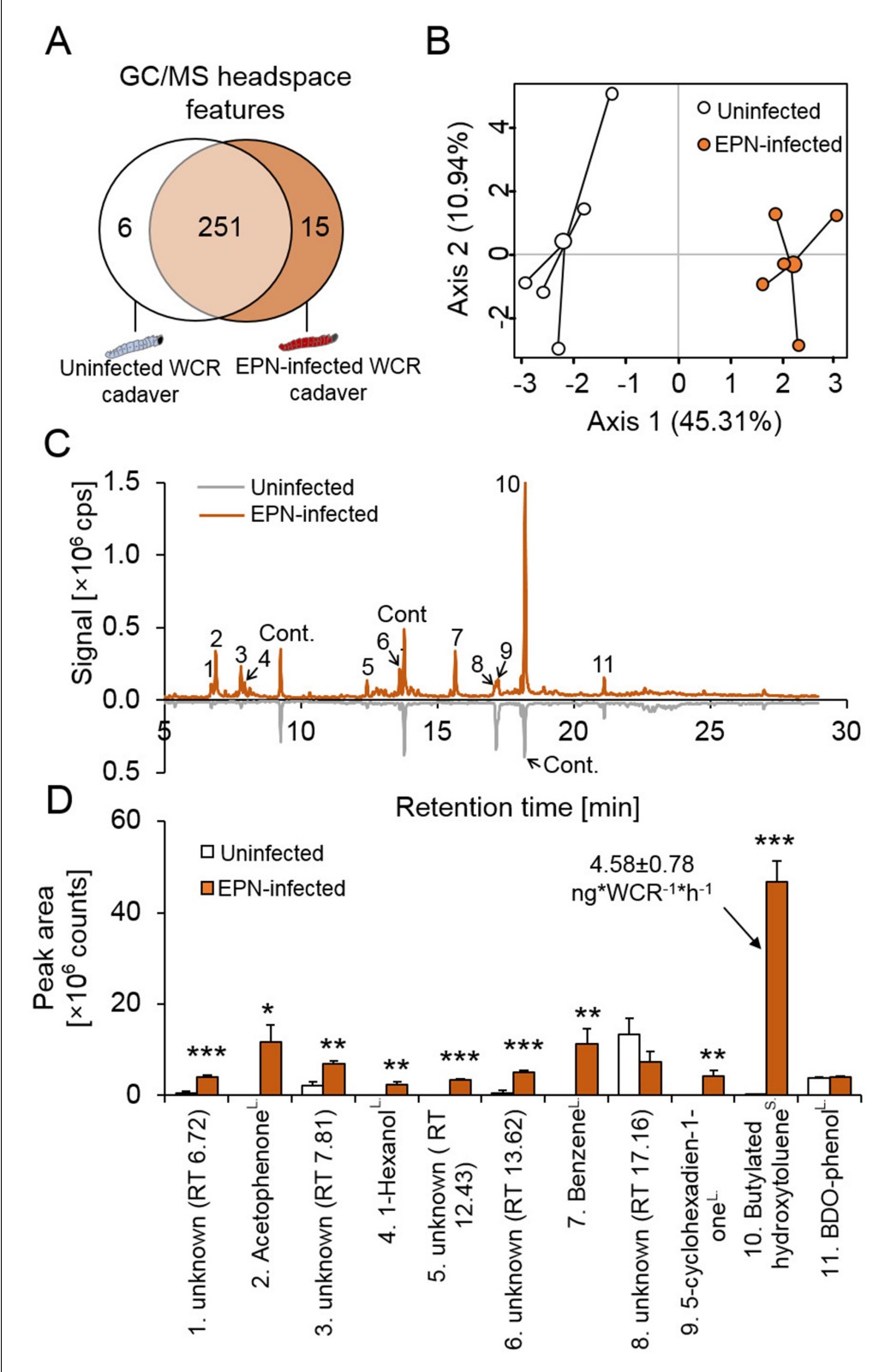

**Figure 3.** Western corn rootworm larvae that are infected by entomopathogenic nematodes release distinct volatile bouquets. (**A**) Venn diagram showing the numbers of overlapping and non-overlapping GC-MS headspace features of uninfected (white) and EPN-infected (brown) western corn rootworm (WCR) cadavers (n = 5). (**B**) Principal component analysis (PCA) of volatile emissions of uninfected (white) and EPN-infected (brown) WCR cadavers (n = 5). (**C**) Representative GC-MS volatile profile of uninfected (white) and EPN-infected WCR larvae (brown). 1, 3, 5, 6, 8: unknown; 2:

*Figure 3 continued on next page*

*Figure 3 continued*

aceptophenone; 4: 1-hexanol; 7: benzene; 9: 5-cyclohexadien-1-one; 10: butylated hydroxytoluene (BHT); 11: 2,6-bis (1,1-dimethylethyl)—4-(1-oxopropyl) phenol; Cont.: contamination. cps: count per second. (D) Volatile peak areas (mean ± SEM) of uninfected (white) and EPN-infected WCR larvae (brown) (n = 5). cps: count per second. [L]: identification based on libraries. [S]: identification based on pure standards. Stars indicate significant differences based on analyis of variance (*: p<0.05, **: p<0.01, ***: p<0.001). Raw data are available in *Figure 3—source data 1*.
DOI: https://doi.org/10.7554/eLife.46668.008
The following source data and figure supplements are available for figure 3:

**Source data 1.** Raw data for *Figure 3*.
DOI: https://doi.org/10.7554/eLife.46668.011
**Figure supplement 1.** Infection by entomopathogenic nematodes does not alter $CO_2$ emissions from western corn rootworm cadavers.
DOI: https://doi.org/10.7554/eLife.46668.009
**Figure supplement 2.** Butylated hydroxytoluene emission is specific to western corn rootworm infection by entomopathogenic nematodes.
DOI: https://doi.org/10.7554/eLife.46668.010

Significantly more WCR larvae were recovered in the vicinity of BHT presence (*Figure 5B*), confirming the attractive effect of BHT in soil. The proportion of EPN-infected WCR larvae was twice higher on the BHT supplemented side (*Figure 5C*). The number of EPN juveniles emerging per larva was similar between the larvae from both sides (*Figure 5D*). Overall, three times more nematodes of the next generation emerged from the BHT side (*Figure 5E*). Thus, BHT enhances the recruitment of healthy WCR larvae, and increases the predation success and total offspring production of EPNs.

As EPNs themselves are also attracted by BHT (*Figure 4C*), we measured whether EPNs from the control side of the arenas may have moved to the BHT side by using a *Galleria melonella* baiting approach (*Orozco et al., 2014*). Application of EPNs to one side of the arena led to *G. melonella* infection of the other side, and the infection rate was slightly increased when BHT was added (*Figure 4—figure supplement 2*). Thus, the observed increase in nematode reproduction in response to cadavers as well as pure BHT is likely to be the result of increased WCR recruitment, increased infection rates and increased EPN recruitment.

## Recruitment of healthy hosts to nematode infested cadavers is widespread and associated with the induction of species-specific volatile profiles

As *H. bacteriophora* is a generalist parasite that can infect many other insect species above and below ground, we investigated whether *H. bacteriophora* infection also increases the recruitment of healthy hosts in other insect species. Five of the seven tested species, including *Diabrotica virgifera* (WCR), *D. balteata*, *Tenebrio molitor*, *Drosophila melanogaster* and *Spodoptera littoralis* were attracted to cadavers of EPN-infected conspecifics (*Figure 6A*). Only the larvae of the wax moth *Galleria melonella* larvae, which are not typically exposed to EPNs in nature, preferred non-infected over infected cadavers. We also tested whether the observed phenomenon is specific to *H. bacteriophora*. We found WCR larvae were attracted to WCR cadavers infested by *H. bacteriophora*, *H. beicherriana*, *H. georgiana* and *S. feltiae* (*Figure 6B*), showing that attraction to cadavers occurs upon infestation with different EPN species. To understand if WCR attraction is specific for EPN-infected WCR cadavers, we measured attraction to cadavers of different insect species. WCR larvae were attracted to all four tested species (*Figure 6C*). On the other hand, when we tested the attraction of these four different insects to EPN-infected WCR cadavers, we found that only the closely related *D. balteata* was attracted to infected WCR cadavers (*Figure 6D*). Thus, attraction of herbivores to infected cadavers is widespread, with the specificity of attraction varying between insect species. To determine whether the attraction of the different insect species to infected conspecifics and the general attraction of WCR to EPN-infected cadavers can be explained by BHT release, we screened volatile emissions of healthy and infected cadavers of all tested insect species. EPN infection significantly altered the volatile bouquets of all insects (*Figure 6—figure supplement 1*). EPN-induced volatiles differed substantially between species. No commonly induced volatiles were found across all species or across the six species that were attracted to infected cadavers. BHT was only released from EPN-infected WCR and *D. melanogaster* cadavers. From these experiments, we conclude that the attraction of healthy conspecifics to EPN-infected cadavers likely involves the emission different sets of attractants.

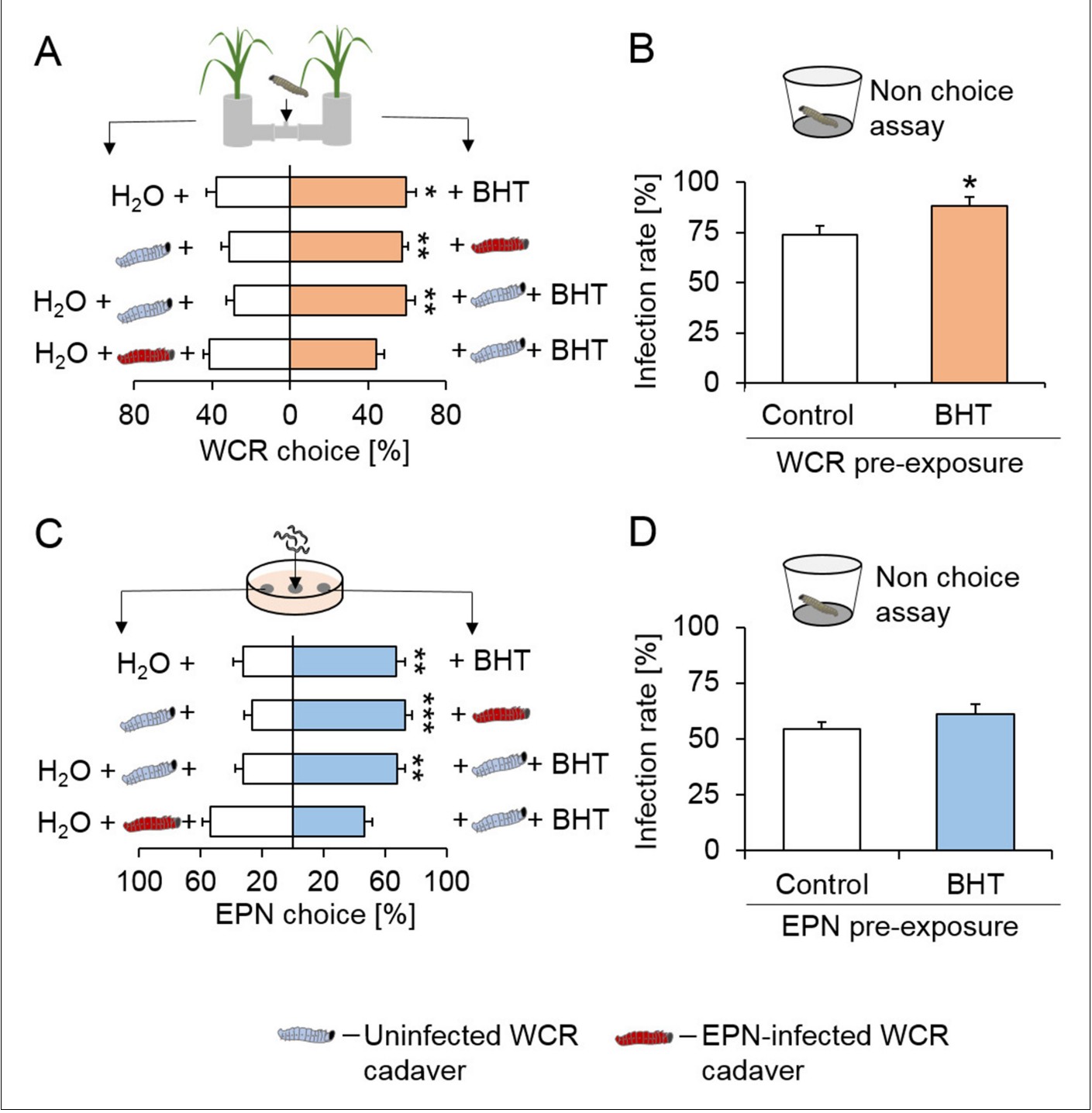

**Figure 4.** Butylated hydroxytoluene attracts herbivores and increases infection by entomopathogenic nematodes. (**A**) Proportions (mean ± SEM) of western corn rootworm (WCR) larvae orienting towards BHT or $H_2O$ (n = 20), uninfected WCR cadavers or cadavers infected by entomopathogenic nematodes (EPNs, n = 10), uninfected WCR cadavers covered with BHT or $H_2O$ (n = 15), EPN-infected WCR cadavers covered with BHT or $H_2O$ (n = 15). (**B**) WCR infection rate (Mean ± SEM) after exposure to BHT or $H_2O$ (n = 10). (**C**) Proportions (Mean ± SEM) of EPNs orienting towards *Diabrotica balteata* exudates complemented with BHT or $H_2O$, uninfected WCR cadavers or cadavers infected by EPNs, uninfected WCR cadavers covered with BHT or $H_2O$, EPN-infected WCR cadavers covered with BHT or $H_2O$ (n = 20). (**D**) WCR infection by EPNs (Mean ± SEM) after pre-incubation of EPNs with BHT or $H_2O$ for 24 hr (n = 15). Stars indicate significant differences based on analysis of variance (*: p<0.05, **: p<0.01, ***: p<0.001). All treatment solutions contained 0.01% ethanol. Raw data are available in *Figure 4—source data 1*.

DOI: https://doi.org/10.7554/eLife.46668.012

*Figure 4 continued on next page*

*Figure 4 continued*

The following source data and figure supplements are available for figure 4:

**Source data 1.** Raw data for *Figure 4*.
DOI: https://doi.org/10.7554/eLife.46668.015
**Figure supplement 1.** Rinsing WCR larvae after exposure to BHT reduces BHT emissions to traces.
DOI: https://doi.org/10.7554/eLife.46668.013
**Figure supplement 2.** Butylated hydroxytoluene attracts entomopathogenic nematodes and increases their predation success in the soil.
DOI: https://doi.org/10.7554/eLife.46668.014

## Discussion

Natural enemies reduce herbivore populations and thereby contribute to the dominance of plants in terrestrial ecosystems and to plant yields in agriculture. Parasites with indirect life cycles are well known to be able to increase their transmission by manipulating host behavior (*Poulin, 2012*; *Poulin and Maure, 2015*), but the prevalence and importance of this phenomenon in parasites with direct life cycles, including herbivore natural enemies, remains largely unexplored. Here, we show that nematode-infection triggers the release of volatiles from the cadavers of herbivorous insects, resulting in the attraction of healthy herbivores, higher infection rates and increased nematode reproduction.

Parasites have developed diverse strategies to manipulate their hosts and thereby increase their fitness (*Moore, 2002*; *Thomas et al., 2005*; *Poulin and Maure, 2015*). Here, we show that entomopathogenic nematodes (EPNs) increase their predation success by inducing the release of volatiles from infected host cadavers. These volatiles attract healthy herbivores and increases nematode infection success. Thus, when nematodes emerge from the exploited cadaver, they will have a higher chance to find healthy hosts, which boosts their chances of survival and reproduction. Other nematodes may also cue in on these volatiles, which may also increase their chance of encountering additional hosts. Together, these phenomena markedly increase top-down control of herbivores in the soil, as shown here for an important agricultural pest, the western corn rootworm. Earlier work shows that EPNs can follow plant volatiles (*Rasmann et al., 2005*), which also serve as aggregation cues to the western corn rootworm (*Robert et al., 2012a*; *Robert et al., 2013*; *Robert et al., 2012b*). The capacity to attract healthy hosts represents a new facet of EPN biology that may explain why EPNs can control the western corn rootworm in the field despite the fact that the insect sequesters plant toxins for self-defense (*Robert et al., 2017*; *Toepfer et al., 2005*; *Toepfer et al., 2008*).

EPN-infected WCR cadavers release a distinct bouquet of volatiles, including butylated hydroxytoluene (BHT). Butylated compounds such as BHT are uncommon in nature, and naturally produced BHT has so far only been documented for a handful of microorganisms (*Babu and Wu, 2008*; *Vikram et al., 2004*). We found that BHT is specifically released from EPN-infected WCR cadavers, and that it is sufficient to elicit WCR behavior similar to EPN-infected cadavers. How BHT is produced in the cadavers requires further study. Digestion of the larvae by symbiotic bacteria of the nematodes is not sufficient to elicit BHT release, suggesting that nematode-specific factors are required. EPNs produce a variety of proteins to overcome and digest their insect hosts (*Lu et al., 2017*), and it is probable that these proteins interact with host-derived metabolites to form BHT. A thorough screen of the rearing and experimental environments used in this study revealed no traces of environmental BHT, which, together with our control experiments, strongly suggests that the BHT we measure is of biological origin. BHT is a radical scavenger that is used as a food additive and synthetic analog of vitamin E (*Burton and Ingold, 1981*). Thus, the production of BHT may have additional benefits to the nematodes, for instance by preserving the herbivore cadavers as they are consumed. From an applied point of view, BHT may represent a cost-effective synthetic substance that could be applied as a bait that attracts the western corn rootworm and its natural enemies.

Parasites typically attract healthy hosts by hijacking adaptive behavioral responses. The flatworm *Leucochloridium paradoxum* for instance modifies the eye stalks of snails to resemble caterpillars, which prompts birds to attack the eyes, thus allowing the flatworm to be transmitted to its primary host (*Wesołowska and Wesołowski, 2014*). Furthermore, the bacterial pathogen *Pseudomonas entomophila* triggers the release of aggregation pheromones from infected *Drosophila melanogaster*, which attracts healthy flies and thus enhances pathogen dispersal (*Keesey et al., 2017*).

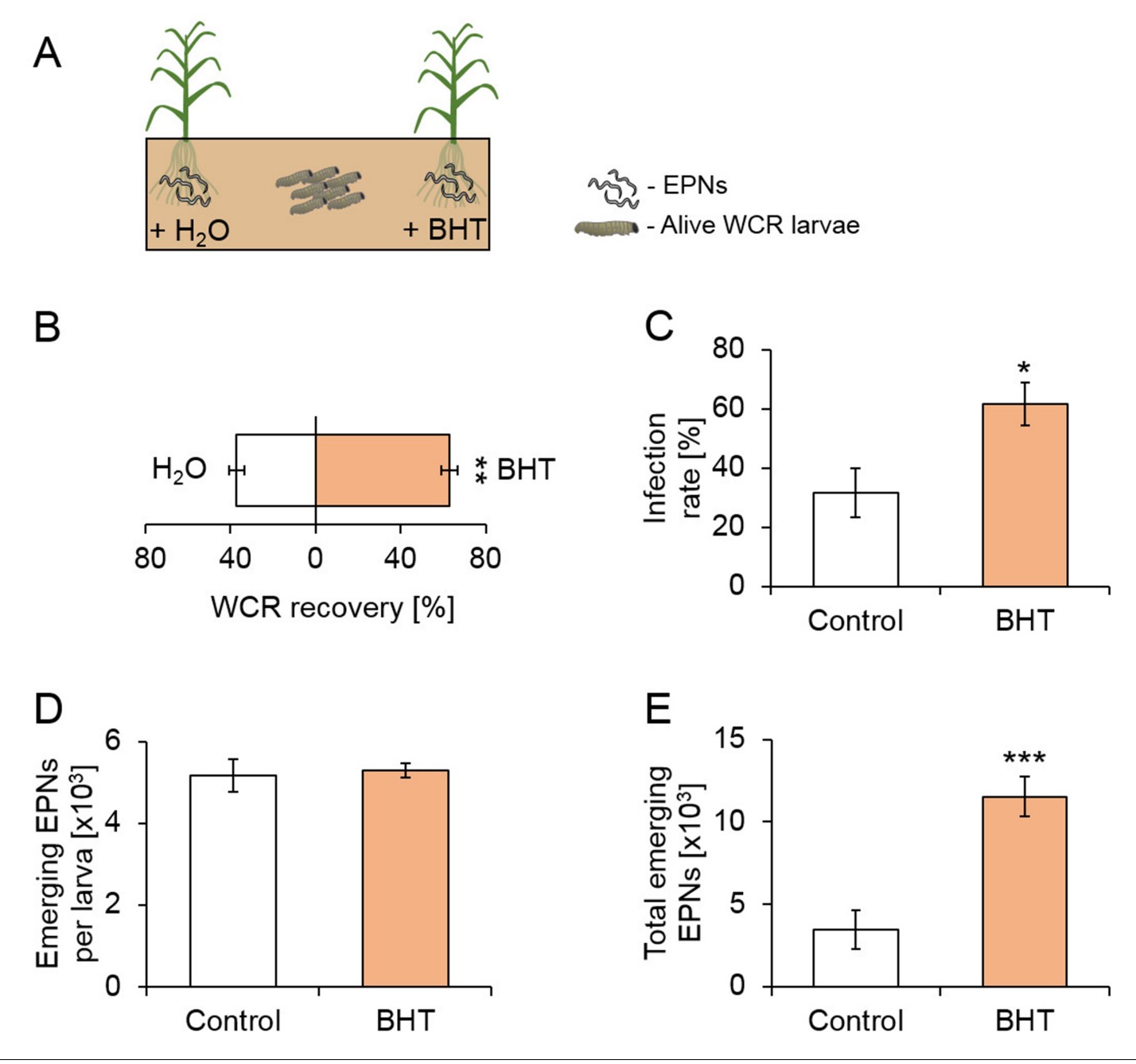

**Figure 5.** Butylated hydroxytoluene increases herbivore recruitment, predation success and reproduction of entomopathogenic nematodes in the soil. (A) Visual representation of experimental setup. Entomopathogenic nematodes (EPNs) were applied on both sides of the arenas, and each side was either watered with butylated hydroxytoluene (BHT) or water ($H_2O$). Eight western corn rootworm (WCR) larvae were then released in the middle and recollected after five days (n = 12). (B) Proportions (Mean ± SEM) of WCR larvae recovered from each side (n = 12). (C) WCR infection rates (Mean ± SEM) on each side (n = 12). (D) Number (Mean ± SEM) of EPN juveniles emerging per WCR larva ($n_{Control}$ = 6, $n_{infected}$ = 12). (E) Total number of EPNs (Mean ± SEM) emerging from the WCR larvae on each side (n = 12). Stars indicate significant differences based on analysis of variance (**: p<0.01, ***: p<0.001). Raw data are available in *Figure 5—source data 1*.

DOI: https://doi.org/10.7554/eLife.46668.016

The following source data is available for figure 5:

**Source data 1.** Raw data for *Figure 5*.

DOI: https://doi.org/10.7554/eLife.46668.017

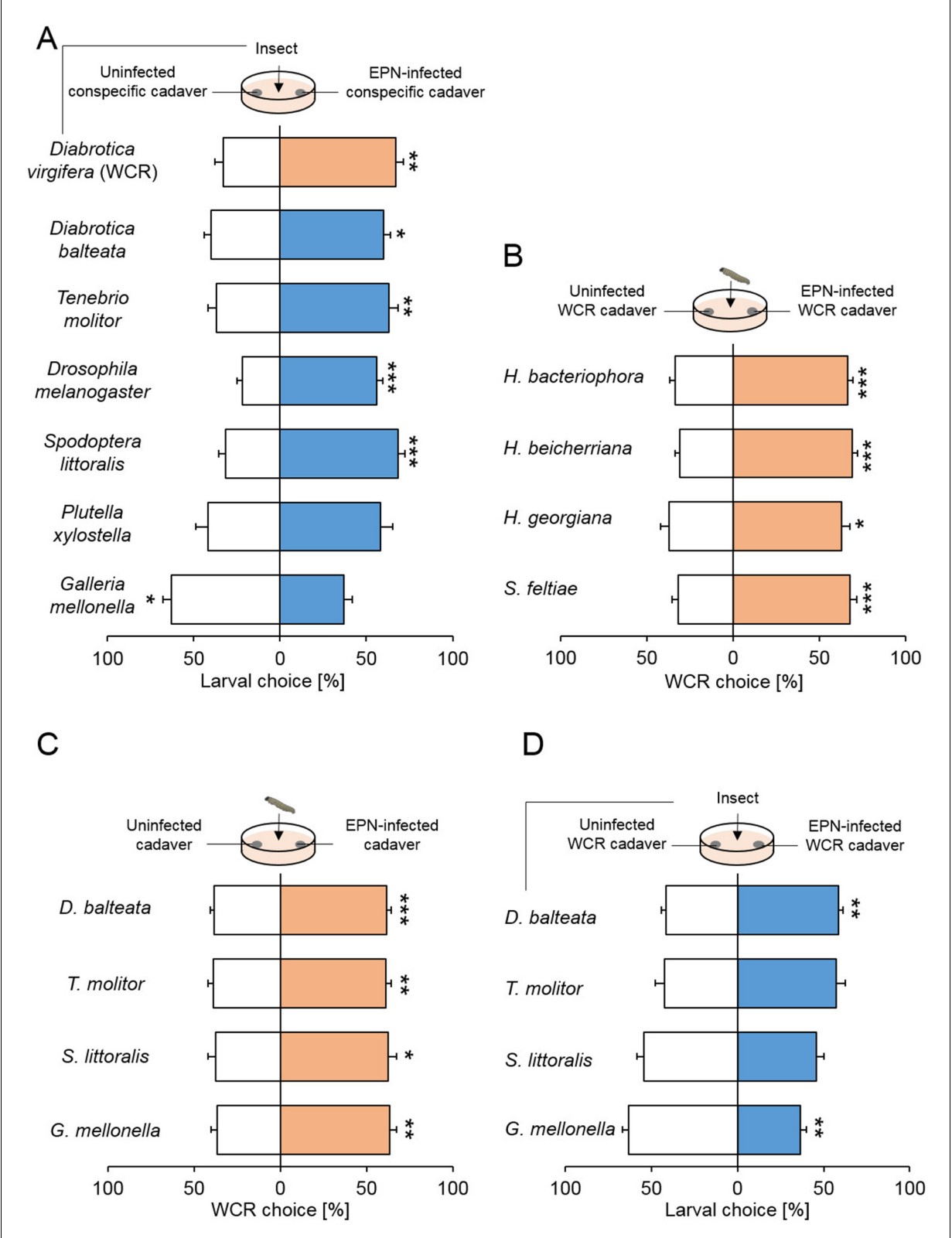

**Figure 6.** Attraction to cadavers infected by entomopathogenic nematodes is widespread. (**A**) Proportions (mean ± SEM) of larvae orienting towards conspecific cadavers that were uninfected or infected by the entomopathogenic nematode (EPNs) *H. bacteriophora*. A total of seven species were tested: *Diabrotica virgifera* (WCR, n = 10), *D. balteata* (n = 19), *Tenebrio molitor* (n = 16), *Drosophila melanogaster* (n = 14), *Spodoptera littoralis* (n = 18), *Plutella xylostella* (n = 16) and *Galleria mellonella* (n = 10). (**B**) Proportions (mean ± SEM) of WCR larvae to their conspecific cadavers that were

*Figure 6 continued*

uninfected or infected by other four species of EPNs. These four EPN species were *H. bacteriophora* (n = 12), *H. beicherriana* (n = 15), *H. georgiana* (n = 13) and *S. feltiae* (n = 15). (C) Proportions (mean ± SEM) of WCR larvae for insect cadavers of different species that were uninfected or infected by *H. bacteriophora*. The four species were *D. balteata* (n = 15), *Tenebrio molitor* (n = 15), *Spodoptera littoralis* (n = 15), and *Galleria mellonella* (n = 15). (D) Proportions (mean ± SEM) of larvae orienting towards *H. bacteriophora* uninfected or infected WCR larvae. Replicate numbers were the same as figure C. Stars indicate significant differences based on analysis of variance (*: p<0.05, **: p<0.01, ***: p<0.001). Raw data are available in *Figure 6— source data 1*.

DOI: https://doi.org/10.7554/eLife.46668.018

The following source data and figure supplement are available for figure 6:

**Source data 1.** Raw data for *Figure 6*.
DOI: https://doi.org/10.7554/eLife.46668.020
**Figure supplement 1.** Attraction of insects to EPN-infected cadavers is widespread.
DOI: https://doi.org/10.7554/eLife.46668.019

We show that EPNs can use volatiles such as BHT to attract healthy rootworm larvae in the soil. Why the rootworm larvae are attracted by the volatiles of infected cadavers is currently unclear. Given that approaching an infected cadaver bears a substantial mortality risk due to the presence of nematode infective juveniles and the suppression of immunity by volatiles such as BHT, it is unlikely that this behavior is adaptive for the herbivore itself. Based on the current literature, it seems more likely that following volatiles such as BHT is beneficial for the rootworm in a different context. Because WCR larvae and adults are attracted to and use certain aromatic compounds for host selection (*Lampman et al., 1987*; *Erb et al., 2015*), one hypothesis is that BHT attracts the larvae either by interacting with the receptors of compounds involved in host location or by mimicking their activity. Such effects were for instance reported for volatile odorants blocking $CO_2$ receptors and responses in fruit flies and mosquitoes (*Turner et al., 2011*; *Turner and Ray, 2009*).

Even though the benefits of attracting healthy rootworms for EPNs seems evident, whether this is a true form of manipulation requires further mechanistic, evolutionary and ecological insights. As *H. bacteriophora* is a generalist with a broad host range, we hypothesized that the nematode should be able to induce attractive volatiles in a wide variety of hosts in order to benefit from this trait. Indeed, nematode-infestation triggered attraction of healthy conspecifics in five out of seven tested insect species and was induced by four different EPN species, suggesting that this phenomenon is widespread and may benefit different nematodes in the presence of different insect host species. Strikingly, WCR larvae were attracted to a variety of different EPN-infected hosts, suggesting that the nematodes may, at least in the case of WCR, also succeed in attracting additional hosts of other species. However, most of the other tested insect species were not attracted to nematode-infested WCR larvae, showing that the attraction to infected cadavers is not universal and, to a certain degree, specific. Our analyses of the volatile blends that are emitted upon infection by the different insects also reveal a high degree of specificity, with each insect producing a different, attractive volatile blend, with little overlap between the different species. It is tempting to speculate that *H. bacteriophora* may have the capacity to adjust the induction of volatiles from the host it invades to maximize the attraction of healthy hosts. Different insect species are attracted by different volatiles (*Müller et al., 2015*; *Gershenzon and Dudareva, 2007*; *Dicke and Baldwin, 2010*), which makes such a mechanism beneficial as host communities vary. A better understanding of the proximal mechanisms of volatile induction by nematode infection and perception by different herbivores would help to shed light on this hypothesis (*Herbison et al., 2018*).

A recent study found that volatiles from EPN-infected insect cadavers increases plant resistance against the Colorado potato beetle (*Helms et al., 2019*). EPN-induced cadaver volatiles can thus affect herbivores directly, by altering their behavior, or indirectly, *via* plant-mediated changes. The different and diverse effects of EPN-induced cadaver volatiles should be better characterized and integrated in future studies to fully understand how the presence of herbivore natural enemies affects plant-herbivore interactions.

In conclusion, this study demonstrates that infection with EPNs triggers the release of volatiles that are attractive to healthy hosts and suppress their nematode resistance, which increases predation success and top-down control of a herbivore pest. The finding that nematode infection increases the recruitment of healthy hosts across different insect species suggests that this

phenomenon is widespread and may contribute to shaping the interactions between insects and their natural enemies in nature and in the context of the biological control of soil-borne insect pests.

# Materials and methods

## Key resources table

| Reagent type (species) or resource | Designation | Source or reference | Identifiers | Additionnal information |
|---|---|---|---|---|
| Biological sample (*Zea mays L.*) | Maize; *Zea mays* | Delley semences et plantes SA, Swizerland | Akku | |
| Biological sample (*Diabrotica virgifera virgifera LeConte*) | Western corn rootworm; WCR | USDA-ARS-NCARL, Brookings, SD, USA | | |
| Biological sample (*Diabrotica balteata LeConte*) | *Diabrotica balteata; D. balteata* | Syngenta Crop Protection AG, Stein, Switzerland | | |
| Biological sample (*Plutella xylostella*) | *Plutella xylostella; P. xylostella* | Syngenta Crop Protection AG, Stein, Switzerland | | |
| Biological sample (*Drosophila melanogaster*) | *Drosophila melanogaster; D. melanogaster* | University of Bern (Bern, Switerland) | | |
| Biological sample (*Spodoptera littoralis*) | *Spodoptera littoralis; S. littoralis* | University of Neuchâtel, Neuchâtel, Switzerland | | |
| Biological sample (*Galleria melonella*) | *Galleria melonella; G. melonella* | Fischereibedarf Wenger, Bern, Swizerland | | https://www.fischen-wenger.ch/ |
| Biological sample (*Tenebrio molitor*) | *Tenebrio molitor; T. molitor* | Fischereibedarf Wenger, Bern, Swizerland | | |
| Biological sample (*Heterorhabditis bacteriophora*) | *Heterorhabditis bacteriophora; H. bacteriophora* | Andermatt Biocontrol, Grossdietwil, Swizerland | Meginem (643C) | https://www.biocontrol.ch/de_bc/meginem-pro |
| Biological sample (*Steinernema feltiae*) | *Steinernema feltiae; S. feltiae* | Andermatt Biocontrol, Grossdietwil, Swizerland | Traunem (1008C) | https://www.biocontrol.ch/de_bc/traunem |
| Biological sample (*H. beicherriana*) | *H. beicherriana* | Hebei Academy of Agricultural and Forestry Science/IPM center of Hebei Province, China | | *Ma et al., 2013* |
| Biological sample (*H. georgiana*) | *H. georgiana* | Own collection from the USA | | |
| Chemical compound, drug | Butylated hydroxytoluene; BHT | Sigma Aldrich | W218405 | https://www.sigmaaldrich.com/catalog/product/aldrich/w218405?lang=de®ion=CH |
| Chemical compound, drug | Ethanol | Sigma Aldrich | 51976 | https://www.sigmaaldrich.com/catalog/product/sial/51976?lang=de®ion=CH |
| Software, algorithm | R 3.2.2 | R Foundation for Statistical Computing, Vienna, Austria | | https://www.r-project.org/ |

*Continued on next page*

*Continued*

| Reagent type (species) or resource | Designation | Source or reference | Identifiers | Additionnal information |
|---|---|---|---|---|
| Software, algorithm | Progenesis QI | informatics package from Waters, MA, USA | | https://www.waters.com/waters/de_CH/Progenesis-QI/nav.htm?cid=134790652&locale=de_CH |
| Software, algorithm | NIST search Mass Spectral Library, NIST 2.2 | National Institute of Standards and Technology Standard Reference Data Program Gaithersburg, USA | | |
| Other | GC-MS | Agilent Technologies (Schweiz) AG, Basel, Switzerland | | |
| Other | SPME fiber | Supelco, USA | | 100 μm polydimethylsiloxane coating |

## Biological resources

Maize plants (*Zea mays* L.; variety 'Akku', Delley semences et plantes SA, Swizerland) were grown in a greenhouse (23 ± 2℃, 60% relative humidity, 16:8 hr L/D, and 250 µmol.m$^{-2}$.s$^{-1}$ light). Twelve-day-old plants were used for all experiments. Western corn rootworm (WCR, *Diabrotica virgifera virgifera* LeConte) eggs were kindly provided by Chad Nielson and Wade French (USDA-ARS-NCARL, Brookings, SD, USA). *Diabrotica balteata* (LeConte) and *Plutella xylostella* eggs were kindly supplied by Oliver Kindler (Syngenta Crop Protection AG, Stein, Switzerland). *Drosophila melanogaster* larvae were obtained from institute of cell biology, University of Bern (Bern, Switerland). *Spodoptera littoralis* eggs were provided by Ted Turlings (University of Neuchâtel, Neuchâtel, Switzerland). *Galleria melonella* and *Tenebrio molitor* larvae were obtained from a commercial vendor (Fischereibedarf Wenger, Bern, CHE). Entomopathogenic nematodes (EPNs) *Heterorhabditis bacteriophora* strain Andermatt and *Steinernema feltiae* strain Andermatt were obtained from a commercial vendor (Andermatt Biocontrol, Grossdietwil, CHE). *H. beicherriana* strain LJ-24 was kindly provided by Juan Ma (Hebei Academy of Agricultural and Forestry Science/IPM center of Hebei Province, China). *H. georgiana* strain S10. All EPNs were reared in *G. mellonella* larvae as described previously (*McMullen and Stock, 2014*). EPN-infected insect larvae were obtained by placing larvae in solo-cups (30 mL cups, Frontier Scientific Services, Inc, USA) containing a 0.5 cm layer of autoclaved moist sand (Selmaterra, Bigler Samen AG, Thun, CHE) and 1000 infective juveniles (IJs) in 500 µL tap water. Uninfected controls were obtained by adding 500 µL tap water to the cups. *H. bacteriophora* IJs were used for all the experiments, unless specified otherwise.

## WCR preference

*Belowground olfactometer experiments.* The impact of EPN infection on WCR behavior was investigated in a series of experiments using dual-choice olfactometers as described previously (*Robert et al., 2012a*). Briefly, different combinations of plants, EPNs and WCR larvae were placed in L-shaped glass pots. One control plant and one treated plant were connected by one glass connector closed by two Teflon connectors containing a fine mesh that prevented the larvae from accessing the plant root system while allowing volatiles to diffuse freely. Five third-instar WCR larvae were added to the central connector. The first choice of the larvae was recorded. Larvae remaining in the central connector longer than 15 min were recorded as 'no choice'. First, WCR larvae were given the choice between a healthy maize plant or a maize plant infested with five WCR larvae and 2000 EPNs. WCR choice was evaluated 0, 24, 48, 72 and 96 hr after infestation. Second, WCR preference 96 hr post infestation was investigated by offering larvae the choice between the following combinations: (i) maize *vs.* maize+EPNs; (ii) maize *vs.* maize+ WCR; (iii) maize *vs.* maize+WCR+EPN and (iv) five WCR larvae enclosed in a filter paper cage next to a healthy plant *vs.* five EPN-infected WCR larvae enclosed in a filter paper cage next to a healthy plant. Infestation of plants with WCR larvae was realized by adding five healthy third instar larvae to the bottom opening of the glass

pots. EPN treated plants were obtained by adding 10 mL EPN solutions (200 IJs/mL with 30 mg EPN medium from Andermatt Biocontrol, CHE) to the bottom opening of the glass pots. Plants without EPNs were obtained by adding 10 mL of 3 mg/mL of EPN medium.

*Petri dish assays.* BHT attractiveness to WCR larvae was assessed using Petri dishes (9 cm diameter, Greiner Bio-One GmbH, Frickenhausen, DE). A 5 mm layer of 1% agarose (m/v, Sigma Aldrich Chemie, CHE) was poured into the dishes. Two to three 5 cm root pieces were placed at the two opposite sides of the dish. Two filter paper (90 mm, Whatman, Sigma Aldrich Chemie, CHE) slices (length = 7 cm, width = 1 cm) were placed in parallel in between the root sections at four cm distance from each other. Five frozen-killed control WCR larvae or EPN-infected WCR larvae (five days post-infection) were placed onto the filter paper slice respectively. BHT complementation was realized by adding 20 ng BHT in 100 µL 0.01% ethanol ($\geq$99.8%, Sigma Aldrich Chemie, CHE) onto the filter paper slice. This dose corresponds to the amount that is released by four infected larvae over an hour (*Figure 3D*). Control slices were imbibed with 100 µL 0.01% ethanol. Five third instar WCR larvae were given the choice between (i) control and BHT slices, (ii) healthy and EPN-infected larvae, (iii) healthy larvae and healthy larvae complemented with BHT and (iv) EPN-infected larvae and healthy larvae complemented with BHT. The choice of WCR larvae was measured by adding five larvae in the center of the dish and recording their positions after 0.5 hr, 1.5 hr, 3 hr and 5 hr. To gain more insights into WCR behavior, this experiment was repeated and WCR behavior towards EPN-infected WCR cadavers (frequency and duration of contact) was continuously recorded over two hours. Using this petri-dish assay, the specificity of the preference response was compared in four experiments. First, the preference of six other insect species including *D. balteata*, *T. molitor*, *D. melanogaster*, *S. littoralis*, *P. xylostella* and *G. mellonella* to healthy and *H. bacteriophora*-infected conspecifics was assessed. Second, the preference of WCR larvae for conspecifics infected with different EPN species, including *H. megidis*, *H. beicherriana*, *H. georgiana* and *S. feltiae*, was tested. Third, the preference of WCR larvae between healthy and *H. bacteriophora*-infected *D. balteata*, *T. molitor*, *S. littoralis* and *G. mellonella* larvae was evaluated. Fourth, the preference of *D. balteata*, *T. molitor*, *S. littoralis* and *G. mellonella* larvae between healthy and *H. bacteriophora*-infected WCR larvae was assessed. The choice of the insects was measured by adding five larvae in the center of the dish and adding food pieces on each side. Food pieced consisted of maize roots (var. Akku) for *D. balteata*, carrot pieces (4 cm$^3$, Bio-carrot, Migros, CHE) for *T. molitor*, artificial diet (1 cm$^3$, Standard Solid Food Diet) for *D. melanogaster*, maize leaves (4 cm$^2$, var. Akku) for *S. littoralis*, cabbage leaves (4 cm$^2$, Bio-cabbage, Migros, CHE) for *P. xylostella*, and artificial diet (1 cm$^3$, Fischereibedarf Wenger, Bern, CHE) for *G. mellonella*. The position of the insects in the dishes was recorded after 0.5 hr, 1.5 hr, 3 hr and 5 hr. Experiments with *T. molitor*, *P. xylostella* and *D. melanogaster* were stopped after 3 hr as considerable amounts of food pieces and/or feces were present in the dishes, possibly biasing any further feeding preference.

## Root consumption by WCR larvae

Root consumption by WCR larvae over time was assessed in belowground olfactometers as described above. Root tissues were collected at 0, 24, 48, 72 and 96 hr after adding five WCR and 2000 EPNs. The difference between the root masses of plant+WCR+EPN complexes and healthy plants was used as a proxy for tissue removal. All collected roots were flash frozen for (*E*)-β-caryophyllene analyses (see section 'Volatile analysis').

## EPN preference

BHT attraction to EPNs was evaluated in petri dish choice assays as described elsewhere (*Robert et al., 2017*). Briefly, a five mm layer of 0.5% agarose (Sigma Aldrich Chemie, CHE) was poured into the petri dishes. To test EPN behavioral response to BHT, exudates from healthy *D. balteata* larvae were complemented with BHT or water (control) and placed in two 5 mm diameter wells along the plate diameter and at 5 cm distance from each other. Exudates from *D. balteata* larvae were collected by rinsing third instar larvae with tap water (50 µL per larva). BHT exudate complementation was performed by adding 20 ng BHT in 0.01% ethanol per well. Control exudates were obtained by adding the equivalent volume of 0.01% ethanol to the wells. BHT complementation of WCR larvae was realized by adding 20 ng BHT in 50 µL 0.01% ethanol onto frozen-killed control WCR larvae. Control larvae were obtained by covering frozen-killed control WCR larvae with 50 µL

0.01% ethanol. A third 5 mm diameter well was made in the center of the dishes to place sixty EPNs suspended in 100 μL water. EPNs were given a choice between: (i) control *vs.* BHT complemented insect exudates, (ii) healthy *vs.* EPN-infected larvae, (iii) healthy larvae *vs.* healthy larvae complemented with BHT and (iv) EPN-infected larvae *vs.* healthy larvae complemented with BHT. The number of EPNs in each side was assessed 24 hr later.

## EPN infectivity and fecundity

EPN infection rates in belowground olfactometers were determined by collecting the larvae at 0, 24, 48, 72 and 96 hr after adding WCR larvae and EPNs. The infection status of the WCR larvae was assessed visually.

The impact of EPN-infected cadaver presence on EPN predation success was tested in soil arenas. Six maize plants were sown in rectangular plastic trays (25 cm *11.5 cm * 9.5 cm, Migros, Bern, CHE), such as two sets of each three plants grew at about 15 cm distance from each other. After 12 days, five cadavers of frozen-killed control larvae were buried on one side, and five cadavers of EPN-infected larvae (4 days post infection) were buried on the second side. EPNs (1500 EPN in 2 mL tap water) were added on each side. Finally, eight WCR larvae were placed in the middle section for four days. After this period, all larvae were collected to record infection rates. Infection rates were only considered when at least two larvae were collected. The number of emerging EPNs per larva was evaluated as a proxy for EPN fecundity. Counting emerging EPNs was performed by placing individual larvae in adapted white traps (*McMullen and Stock, 2014*). Briefly, the white trap consisted of 1.5 mL Eppendorf lid (Sarstedt AG and Co., Germany) placed upside down in a solo cup. The lid was covered with 2.5 cm diameter filter paper. Tap water was placed around the lid. Each larva was placed onto the filter paper and emerging EPNs could reach the water. The number of freshly emerged EPNs was counted 15 days after the emergence of the first EPNs. All larvae collected from one side of one arena were considered as pseudo-replicates and the number of emerging EPNs was averaged within each side for analyses.

The direct and indirect (WCR-mediated) impact of BHT on EPN infectivity was evaluated in three experiments. First, the effect of BHT exposure on the resistance of WCR towards EPNs was tested by adding 50 μL of 0.4 ng/μL BHT in 0.01% ethanol or 50 μL 0.01% ethanol only on a slice of filter paper in a solo cup containing five third instar WCR. One day later, all the larvae were washed with 100% ethanol and tap water. WCR larvae were placed in new solo cups and 200 EPNs in 500 μL tap water were added. The resulting infection rate was recorded five days later.

Second, the effect of BHT exposure on EPN infectivity was tested by incubating EPNs in 0.2 ng BHT/μL in 0.01% ethanol for 24 hr. After incubation, EPNs were washed twice with ethanol and tap water and then 500 EPNs were added into solo cups containing five third-instar WCR larvae. EPNs incubated in 0.01% ethanol were used as controls. The infection rate was recorded 5 days later.

Third, the impact of BHT release by EPN-infected cadavers on EPN predation success was tested in soil arenas as described above. Six maize plants were grown in rectangular plastic trays. After 12 days, 1500 EPNs in 2 mL 0.01% ethanol in water containing 40 ng of BHT were added on one side, while 1500 EPNs in 2 mL 0.01% ethanol in water only were added on the other side. Eight WCR larvae were placed in the middle section. Four days later, all larvae were collected, and the infection rate and number of emerging EPNs per larva was recorded as described above.

## Volatile analysis

Plant and insect volatile emissions were determined by solid-phase micro-extraction-gas chromatography-mass spectrometry (SPME-GC-MS) as previously described (*Robert et al., 2012a*). Briefly, root tissues were ground in liquid nitrogen to a fine powder, and 100 mg portions were weighed into 20 mL glass vials (Gerstel, Germany) for analysis as described below. Insect larvae were all flash frozen and five larvae where placed into a 20 mL glass vial. Gas chromatography analyses were performed using an Agilent 7820A GC interfaced with an Agilent 5977E MSD following established protocols (*Erb et al., 2011*) with a few modifications. Specifically, the SPME fiber (100 μm polydimethylsiloxane coating, Supelco, USA) was inserted into the vial for 30 min. The fiber was desorbed at 220°C for 2 min. The column temperature was initially set at 60°C for 1 min and increase to 250°C at a speed of 5 °C min$^{-1}$ for plant samples, and 200°C for insect samples. The resulting GC-MS chromatograms were processed with Progenesis QI (informatics package from Waters, MA,

USA). Compound identification was realized using the NIST search 2.2 Mass Spectral Library and pure compound standards. The mass spectrum and retention time of synthetic butylated hydroxytoluene (Sigma Aldrich Chemie, CHE) matched those of the putative BHT peak in natural samples. BHT release was quantified using a standard curve of the pure compound in 0.01% ethanol (v/v). Possible contamination with BHT was evaluated by running blanks and performing volatile analyses on all materials used during experiments, including filter papers, petri dishes, solo-cups, Eppendorf tubes, white trap cups, rearing boxes (empty and in use), sand, soil, tap water, and distilled water.

### Data analysis

Preference data were analyzed by comparing the average difference between the proportion of WCR larvae or EPNs choosing control and treated sides to the null hypothesis $H_0 = 0$ using analysis of variance (one-sample t-tests) (*Erb et al., 2015*). This approach is more conservative that other approaches such as generalized linear mixed models (GLMMs) that have been used before to analyze choice experiments (*Hu et al., 2018*). All significant effects reported were confirmed to be statistically significant when analyzed by GLMMs. The other experiments were analyzed by analysis of variance. For multiple comparisons, Least Squares Means (LSMeans) and FDR-corrected post hoc tests (*Benjamini and Hochberg, 1995*) were used. All analyses were carried out using R 3.2.2 (R Foundation for Statistical Computing, Vienna, Austria).

## Acknowledgements

We thank Marc Pfander for helping with data analysis, David Ermacora for rearing EPNs, Sarah Ettlin for helping with experiments, the gardeners of the Institute of Plant Sciences for their support with growing maize plants, and the University of Bern for providing research infrastructure. We are also grateful to the reviewers for the constructive and valuable comments on a previous version of this manuscript.

## Additional information

### Funding

No external funding was received for this work.

### Author contributions

Xi Zhang, Data curation, Formal analysis, Methodology, Writing—original draft, Writing—review and editing; Ricardo AR Machado, Cong Van Doan, Carla CM Arce, Lingfei Hu, Data curation, Formal analysis, Validation, Writing—review and editing; Christelle AM Robert, Conceptualization, Data curation, Formal analysis, Supervision, Validation, Investigation, Visualization, Methodology, Writing—original draft, Writing—review and editing

### Author ORCIDs

Xi Zhang (ID) https://orcid.org/0000-0002-8552-297X
Ricardo AR Machado (ID) https://orcid.org/0000-0002-7624-1105
Cong Van Doan (ID) https://orcid.org/0000-0001-9189-1301
Carla CM Arce (ID) https://orcid.org/0000-0002-1713-6970
Lingfei Hu (ID) https://orcid.org/0000-0002-7791-9440
Christelle AM Robert (ID) https://orcid.org/0000-0003-3415-2371

### Decision letter and Author response

Decision letter https://doi.org/10.7554/eLife.46668.023
Author response https://doi.org/10.7554/eLife.46668.024

## Additional files

### Supplementary files

• Transparent reporting form

DOI: https://doi.org/10.7554/eLife.46668.021

**Data availability**
Source data files have been provided for all main figures.

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
