## [Decision Letter]

Thank you for submitting your article "Entomopathogenic nematodes increase predation success by inducing cadaver volatiles that attract healthy herbivores" for consideration by *eLife*. Your article has been reviewed by three peer reviewers, and the evaluation has been overseen by a Reviewing Editor and Detlef Weigel as the Senior Editor. The following individuals involved in review of your submission have agreed to reveal their identity: Lukasz Stelinski (Reviewer #1); Selcuk Hazir (Reviewer #3).

The reviewers have discussed the reviews with one another and the Reviewing Editor has drafted this decision to help you prepare a revised submission.

The manuscript of Zhang and colleagues makes important progress in understanding the role of volatile signals that mediate in (soil-borne) tritrophic interactions. The study shows that the third trophic level, that is entomopathogenic nematodes, can increase their infection efficacy by manipulating host behavior. It demonstrates that these nematodes cause the release of specific volatiles from infected herbivore cadavers to attract more herbivore hosts. The authors were able to identify a volatile cue of infected herbivore carcasses that attract healthy herbivores, thereby increasing nematode infection rates.

Although we agreed that the data is intriguing and deserves publication, we did not agree with one of the conclusions: in subsection “Butylated hydroxytoluene emitted by nematode-infected cadavers attracts healthy hosts and renders them more susceptible to nematode attack” and the final paragraph of the Discussion section you claim that BHT emitted by nematode-infected cadavers "renders healthy hosts more susceptible to nematode attack". We judge that this claim is insufficiently supported by evidence. We also believe it is more farfetched than the alternative hypothesis that the pre-incubation with BHT simply makes larvae more attractive and thereby stimulates infection. This could be investigated by comparing the nematode infection rate or percentage of BHT treated and untreated larvae in a non-choice assay. Also an analysis of the headspace of BHT treated and untreated larvae may help here to explore the different scenarios. However, such data were not presented.

Therefore we ask for the following:

1) The claim that BHT suppresses larval resistance to nematodes should be removed from the manuscript. It can be addressed as a hypothesis in the Discussion provided the alternative scenario of increased infection caused by increased attractiveness of larvae when treated with BHT is addressed as well.

In subsection “Petri dish assays” you describe petri dish choice assays. We noticed that the WCR data was reported in a different manner (number and duration of visits) than the data for the other species. The description of the tests for the other species is also very superficial, while these results are an important part of your story. Therefore we ask you:

2) To describe the methodology in more detail to make clear in which aspects the WCR tests and the other tests differed. At least:

a) explain in the manuscript why you decided to report on% choice instead of number and duration;

b) explain in the Materials and methods section how/when the% choice was determined e.g. at which time points.

3) Indicate in the Materials and methods section if – and if so, how – additional larval attractants were used such the maize roots for WCR.

---

## [Author Response]

The manuscript of Zhang and colleagues makes important progress in understanding the role of volatile signals that mediate in (soil-borne) tritrophic interactions. The study shows that the third trophic level, that is entomopathogenic nematodes, can increase their infection efficacy by manipulating host behavior. It demonstrates that these nematodes cause the release of specific volatiles from infected herbivore cadavers to attract more herbivore hosts. The authors were able to identify a volatile cue of infected herbivore carcasses that attract healthy herbivores, thereby increasing nematode infection rates.Although we agreed that the data is intriguing and deserves publication, we did not agree with one of the conclusions: in subsection “Butylated hydroxytoluene emitted by nematode-infected cadavers attracts healthy hosts and renders them more susceptible to nematode attack” and the final paragraph of the Discussion section you claim that BHT emitted by nematode-infected cadavers "renders healthy hosts more susceptible to nematode attack". We judge that this claim is insufficiently supported by evidence. We also believe it is more farfetched than the alternative hypothesis that the pre-incubation with BHT simply makes larvae more attractive and thereby stimulates infection. This could be investigated by comparing the nematode infection rate or percentage of BHT treated and untreated larvae in a non-choice assay. Also an analysis of the headspace of BHT treated and untreated larvae may help here to explore the different scenarios. However, such data were not presented.

First, we agree that the claim of BHT-induced susceptibility was too strong. We found that exposing WCR larvae to BHT increase their infection rate by nematodes by a mechanism different than through higher nematode attraction (<0.001 ng.WCR^-1^.h^-1^ BHT is emitted by BHT-exposed and rinsed larvae and those amounts did not trigger a preference behavior for nematodes but still increased their infection rates). Yet, we agree with the fact that we can’t prove that BHT exposure increases susceptibility, but may possibly result from a semi direct effect on nematodes in their hosts. We therefore added the data mentioned above and toned down our interpretation.

Second, we developed the Materials and methods section concerning insect preference assays and provided with schemes and pictures of the used designs. Yet, we could not use the same subtitles for the Materials and methods and Result sections, as most Materials and methods are used for Figures 1 and 2, and therefore lead to several empty method sub-sections (only “see above”). To help the reader understand which design was used, we added schemes on the figures and clearly mention them in the text.

Therefore we ask for the following:1) The claim that BHT suppresses larval resistance to nematodes should be removed from the manuscript. It can be addressed as a hypothesis in the Discussion provided the alternative scenario of increased infection caused by increased attractiveness of larvae when treated with BHT is addressed as well.

We thank you for this comment. We agree that our claim may not have been sufficiently supported by the data presented in the manuscript. First, we would like to clarify that Figure 4B presents the data for a non-choice experiment. Second, we had assessed BHT emissions from the larvae after rinsage and found that rinsing WCR larvae that were exposed to BHT is enough to remove most BHT from their surface. The chromatograms are now added as figure supplement (Figure 4—figure supplement 1A). As traces of BHT remained (<0.001 ng.WCR^-1^.h^-1^) we conducted a nematode choice experiment between control and BHT-exposed and rinsed WCR larvae and found no nematode preference. These results are also now included in the manuscript (Figure 4—figure supplement 1B). Finally, we can’t exclude that WCR sequesters BHT in their body, which could have a direct effect on nematodes. We therefore toned down our claim about induced susceptibility in the Results/Discussion.

In subsection “Petri dish assays” you describe petri dish choice assays. We noticed that the WCR data was reported in a different manner (number and duration of visits) than the data for the other species. The description of the tests for the other species is also very superficial, while these results are an important part of your story. Therefore we ask you:2) To describe the methodology in more detail to make clear in which aspects the WCR tests and the other tests differed. At least:a) explain in the manuscript why you decided to report on% choice instead of number and duration;b) explain in the Materials and methods section how/when the% choice was determined e.g. at which time points.

We thank you for these remarks.

2a) WCR preference was also assessed in proportion of WCR on each sides of the petri dishes. The data were presented in Supplementary Figure 1 but are now included in Figure 1 to increase consistency throughout the figures. Because of the interesting pattern we observed (more% of WCR larvae were found on the side of infected conspecifics), we repeated the experiment to get more insights into the WCR behavior (number of visits/duration). This experiment was only conducted with WCR larvae. This is now clarified in the manuscript.

2b) Done. We developed the method regarding the different insect assays.

3) Indicate in the Materials and methods section if – and if so, how – additional larval attractants were used such the maize roots for WCR.

Done.